# Metabolome-Based Classification of Snake Venoms by Bioinformatic Tools

**DOI:** 10.3390/toxins15020161

**Published:** 2023-02-15

**Authors:** Luis L. Alonso, Julien Slagboom, Nicholas R. Casewell, Saer Samanipour, Jeroen Kool

**Affiliations:** 1Division of BioAnalytical Chemistry, Amsterdam Institute of Molecular and Life Sciences, Vrije Universiteit Amsterdam, De Boelelaan 1085, 1081 HV Amsterdam, The Netherlands; 2Centre for Analytical Sciences Amsterdam (CASA), 1012 WX Amsterdam, The Netherlands; 3Centre for Snakebite Research and Interventions, Liverpool School of Tropical Medicine, Pembroke Place, Liverpool L3 5QA, UK; 4Van ‘t Hof Institute for Molecular Sciences, University of Amsterdam, Science Park 904, 1098 XH Amsterdam, The Netherlands

**Keywords:** venom variation, metabolomics, data analysis, script-controlled peak integration

## Abstract

Snakebite is considered a neglected tropical disease, and it is one of the most intricate ones. The variability found in snake venom is what makes it immensely complex to study. These variations are present both in the big and the small molecules found in snake venom. This study focused on examining the variability found in the venom’s small molecules (i.e., mass range of 100–1000 Da) between two main families of venomous snakes—Elapidae and Viperidae—managing to create a model able to classify unknown samples by means of specific features, which can be extracted from their LC–MS data and output in a comprehensive list. The developed model also allowed further insight into the composition of snake venom by highlighting the most relevant metabolites of each group by clustering similarly composed venoms. The model was created by means of support vector machines and used 20 features, which were merged into 10 principal components. All samples from the first and second validation data subsets were correctly classified. Biological hypotheses relevant to the variation regarding the metabolites that were identified are also given.

## 1. Introduction

Neglected tropical diseases (NTDs)—such as dengue and leishmaniasis—are a group of diseases that mostly afflict populations found in low- and middle-income countries of the tropics and thus have not received as much attention as other diseases [1]. Within NTDs, snakebite stands out for its high mortality rate, although it was only recently accepted as a Category A neglected tropical disease by the WHO in 2017 [2]. Snakebite results in more than 1.8 million envenomings and 125,000 deaths annually [3]. Although snakebite envenoming can be found in all inhabited continents, it is mostly a public health concern in tropical and sub-tropical areas in developing countries, and in part, this is due to the lack of health system resources found within these countries.

Snake venoms are an intricate mixture of bioactive compounds, and their composition varies extensively between and within snake species. They primarily comprise small molecules—such as neurotransmitters or polyamines—salts, metals and proteins, the latter being the most abundant type of biomolecule found. Although multiple types of proteins can be found within snake venom [4,5,6], four stand out as generally being the most important ones based on toxicity and relative abundance [7,8]: three-finger toxins (3FTxs), phospholipases (PLA_2_s), snake venom metalloproteases (SVMPs) and snake venom serine proteases (SVSPs).

In addition to these proteinaceous toxins, earlier studies have also demonstrated the presence of other smaller molecules (<1000 Da) in snake venoms [9,10,11]. For a number of these molecules—or metabolites—such as peptides, amines or hydrocarbons, it was shown that they potentially play a role in aiding [8] or inhibiting [12] protein toxins within the venom. However, these small molecules have not been thoroughly studied due to proteins being the main and most recognizable cause of toxicity of the venom. Nevertheless, they could be of significant help when trying to study the inter- and intra-group variability of venoms and when aiming at gaining better holistic understanding of venom functioning.

Independent of their size, most of the toxins within snake venoms can perturb homeostasis of the human body, and the pathologies associated with snake venoms can be summarized in three main groups: hemotoxic, cytotoxic or neurotoxic [13]. Variability in the composition of venoms can be found both between different taxa [4] and within the same taxonomic unit [14,15,16].

This variability is not only found in snake venom proteinaceous toxins. Small molecules can also be found within snake venoms at different concentrations depending on taxonomic differences of the snakes and thus could also be used for classifying snake venoms. The purpose of these molecules within the venoms is not fully understood yet, partially because of the broad variety of different metabolites found, each of them potentially having one—or several—different functions [8]. Additionally, a generalized classification of the said metabolites has not been reached yet—probably due to the lack of information regarding these molecules. One of the purposes of this research will be looking at classification of small molecules in venoms depending on the source of the said metabolites, which can be useful for the classification of snake venoms but also for understanding the relevance of small molecules in this biological matrix by considering how they might be involved in, for example, human envenoming, maintaining venom gland integrity and ecological relationships between snake and prey. Three groups stand out: endogenous metabolites produced by the venom gland, relevant to the venom activity; endogenous metabolites unrelated to venom, generated through metabolic pathways separate from those associated with venom; and exogenous metabolites coming from the snake’s environment.

The classification of snakes that this study aims for must be based on species, which are somewhat unique to the snake’s taxonomic group. This would mean that, by creating the said model, we are not only developing a way of classifying snake venoms—and as such, the snakes from which the venoms were derived—but also a workflow for untargeted studies of the chemical species able to define each taxonomic group. Because the said species must be found in only that group for us to be able to classify the venom based on the said molecules, the biological and evolutionary reasons for those species to be found in the venoms could be inferred and further studied. For this purpose, a comprehensive list containing the *m*/*z*-values and retention times for all features found in the samples is automatically generated and included in “Appendix A: Level of each metabolite” in the Appendix A.

Within the order Squamata, two families stand out as the most dangerous ones: Elapidae and Viperidae [5]. The Viperidae are mainly known for their hemotoxicity [6], while Elapidae snakebite manifests through systemic neurotoxic and cytotoxic illnesses [17]. As can be seen in Figure 1, *Dendroaspis* and *Naja* are part of the Elapidae family, whereas *Crotalus* and *Bothrops* are part of the Viperidae family. Within the *Naja* genus, spitting and non-spitting cobras can be recognized, the first one being able to spit out venom at attackers. Thus, their venom is specialized in causing more pain, which is achieved by upregulating the PLA_2_s toxins [18]. This taxonomic distribution can be found in Figure 1. No subclade was presented further than Asian *Naja* due to the lack of representation from non-spitting African cobras in the model.

To analyze the venoms of different species and focus on their variability, various statistical tools can be used. Principal component analysis (PCA) is one of the most important techniques used in exploratory data analysis. It was developed in 1901 by Karl Pearson [19], and it involves the projection of a dataset into a series of vectors that hold information about the variance found within the said dataset.

After PCA, other statistical techniques can be used to group the data and create a model able to classify samples based on the said groups. If a proper model is built, the family prediction of a snake based solely on the components of its venom can be produced.

This model could be applied in a high-throughput fashion for recognition and classification of snake venoms using data from liquid chromatography–mass spectrometry (LC–MS) analyses of venoms as input, followed by automated feature extraction from the mass spectra to, finally, introduction of the class-defining features into the model. This would eventually output—within certain confidence intervals—the taxon of the snake.

The automated extraction of data from the liquid chromatography–mass spectrometry (LC–MS) analyses, which is always performed when following the analytical workflow, also results in the creation of a comprehensive list containing all *m*/*z*-values and retention times of all compounds detected in each venom within the mass range used for detection (i.e., 100 to 1000 in our case). Thus, by following the analytical workflow, two main results are obtained: a classifier able to predict the family of the snake (in this case either Elapidae or Viperidae) and a comprehensive list containing the processed analytical data on all small molecules found for each venom.

The bioinformatic tools in combination with the analytical approach presented in this study can help better explain why some metabolites are found more commonly in some taxa and could indicate the evolutionary origin and function of the said small molecules.

## 2. Results

The overall workflow started with the LC–MS (Liquid Chromatography—Mass spectrometry) analysis of the small molecules in 50 different venoms—in duplicate—followed by extraction and alignment of the features found (i.e., all detected peaks in MS and their corresponding properties, such as *m*/*z*-value and retention time) for all the analyses. By doing this, a coherent matrix containing all samples (i.e., collection of features per venom and their intensity) is generated; the columns of the matrix are the features found, and the rows are the samples. This comprehensive list can be found in “Appendix A: Level of each metabolite” in the Appendix A.

Afterward, the measured samples were divided in three subsets in order to perform cross-validation of the model. After the validity of the model was proven, the model was looked at to check what features were recognized as relevant, and their concentrations in each venom were then investigated. This step was performed in conjunction with the identification of several metabolites associated with these relevant features for the model by means of the acquired MS/MS spectra of these metabolites or by standard addition when a metabolite was commercially available.

A graphic depiction of this workflow is included in [Fig toxins-15-00161-ch001]. 

### 2.1. Feature Extraction

To check for repeatability of the analyses, the venom of *Naja siamensis* was spiked with a mixture of standards—specified in Appendix A—and analyzed several times before the analysis of all venoms was performed. This venom was chosen arbitrarily, as the same function could be attained by any other venom. As can be seen in “Appendix A—Analysis of repeatability” in the Appendix A, the retention times of the features were sufficiently repeatable for the bioinformatic tool to extract the features and to recognize the same ones among the different repetitions—even if the retention times shifted slightly—meaning that the method was repeatable. The mixture of standards added to this venom was used to investigate whether normalization of the intensity of all features found by an internal standard would render more reproducible results. Thus, the intensities of the signals of these standards were used to normalize the intensities extracted from the features found by means of recalculating all the intensities as a percentage relative to the chosen standard’s signal intensity for all standards included separately. The highest repeatability was found when the intensities extracted were not normalized. This is probably because all intensities were normalized using a single standard at a time, and internal standards should preferably be similar to the compounds that are being standardized, which was impossible for this study, since none of the compounds measured were known at the point of measurement and of data processing. Additionally, taking into account the complexity of snake venom, no analyte can be similar to all of the metabolites found, meaning that no single internal standard could be used. Thus, no standardization was performed on the data. However, one of the standards (d9-Caffeine) was included in all the measurements to check for any issues during the analyses. By monitoring this standard in all measurements, the stability of the analyses could be investigated, from which it was found that the measurements were performed robustly.

After testing the repeatability of the analyses, the 50 different snake venoms (5 *Dendroaspis*, 15 *Naja*—divided into 5 African spitting cobras, 5 African non-spitting cobras, 4 Asian non-spitting cobras and 1 Asian spitting cobra—27 *Crotalus* and 3 *Bothrops*) were analyzed in duplicate by LC–MS in the order specified in “Appendix A: Order of the injections” in the Appendix A. The MS1 data from each analysis were exported and converted into .mzXML by MSConvert and run through the SAFD bioanalytical tool, which extracts the features and their intensities from the .mzXML files, in conjunction with “Computer code S1” of the Appendix A, which aligns the features between samples, allowing us to obtain a coherent list containing all features and their retention times, *m*/*z*-values and intensities for each sample.

Afterward, all samples were divided in three subsets: the model subset, the first validation subset and the second validation subset. The model subset was used to generate a model that would undergo the different validation tests. The first validation subset was used to test the optimal number of PCs used to project the samples onto. The second validation subset was used to test the validity of the model created after the number of PCs had been chosen.

### 2.2. Model Building

The model we developed to classify snakes based on the metabolome of their venoms was built in four steps: pretreatment of the data, PCA, jackknifing and validation.

The data were pretreated to enhance the reproducibility of the method by only considering the features that would appear in both repetitions of a venom when their signals were at least three times the intensity of those features when they were found in Mili-Q Water, which was used as a (blank) sample. Additionally, a variable called “Rep” was defined, which would allow a feature to be taken into account only when it was found across a “Rep” number of samples. Finally, the data were mean centered and autoscaled. By doing this, a clean dataset focused in the variability of each feature was obtained.

PCA is an explorative matrix decomposition method, which allows for dimensionality reduction [20]. It also generates new variables (principal components, PCs), which are composed of the observed variables. The importance of each observed variable in each PC is defined by the loadings matrix, which, when the observed variables are projected onto it, leads to the scores matrix. The scores matrix is thus the projection of the original dataset onto the principal components, allowing for easy multivariate representation of the data within two or three dimensions. More information on this is given in “Appendix A—Further information regarding Principal Component Analysis” in Appendix A.

Leave one out—or *jackknifing*—is a resampling-without-replacement method introduced by Maurice Quenouille in 1949 [21]. It consists of extracting part of the samples with which the model is built and rebuilding the model without the said samples. If performed several times, it allows for the creation of numerous models, depending on how many samples are extracted and what samples they are. In this case, it was used to optimize the Rep variable. For each possible value of Rep, four samples were extracted from each family—Elapidae and Viperidae—and the models were built. Then, the eight samples extracted were projected onto the loadings to obtain their scores and to be classified. This was repeated 1000 times per Rep value to check the robustness of the model. This way of validating a model is called cross-validation, and it is useful when trying to optimize certain variables within the same dataset, as we do not need to obtain copious amounts of data. Additionally, it avoids bias in the creation of the model due to virtually using all the data as samples to be classified.

As mentioned, two validations were performed with data subsets, which were not part of the model-building data subset. The first validation subset was used to optimize the number of PCs to be considered within the clean PCA model (built with the clean dataset), thus creating the optimal model. The second validation subset was used to validate the final model and check its ability to classify unknown samples.

### 2.3. Classification

Some statistical techniques can be used to group the data and create a model able to classify samples based on the said groups. There are several tools that can be used for this goal. For this case study, two will be tested: one probabilistic and one binary linear.

Proximity classification after *k*-means clustering. Introduced by MacQueen in 1967 [22], the clustering method consists of analyzing the difference between the individual scores’ sample vectors within the PCA model and the mean vector of each group or cluster. Thus, the Euclidean distance between each nth dimensional score and each cluster center is calculated.

To perform the classification, these two values are summed, and an algorithm is written, so that the distance to each center is divided by the said sum and multiplied by 100. This value is recorded as the first percentage, and the other percentage is calculated by subtracting the said number from 100. Thus, two percentages are the outcome. Because *k*-means clustering is based on proximity to the cluster center, the highest percentage was chosen as the indicator of the class of the snake family. These percentages are classified in five groups, which represent how certain the algorithm is about the samples coming from the specified cluster based on the indicator mentioned—the highest percentage. This classification is specified in Table 1.

Support vector machines (SVMs). Introduced in 1992 by Boser, Guyon and Vapnik [23], SVMs work by generating a hyperplane, which separates the two classes of data. The samples will be classified as either of the groups depending on which side of the hyperplane they lay on. The SVM chosen during research was a linear Kernel with a hard margin.

### 2.4. Workflow

#### 2.4.1. Model That Used *k*-Means Clustering

After applying PCA, the *jackknifing* step revealed that the appropriate value for the Rep variable was 36, because in 72 of the 100 iterations, all the samples that had been previously extracted were correctly classified. This was the highest number of correctly assigned sub-models. This led to the model being created with seven variables.

The first validation revealed that the number of PCs that could be used to correctly assign the first validation subset was every number in between three and seven. However, if the number of PCs is the same as the number of variables, there is an overfitting issue. Thus, according to the workflow explained in “Appendix A—Further information regarding *k*-means clustering modelling” in Appendix A, the number of PCs chosen was four, as adding more PCs would lead to overfitting of the data [24].

For the second step of the validation, the second validation subset was projected onto the principal components and classified based on Table 1.

#### 2.4.2. Model That Used Support Vector Machines

The lines of code that run the model building can be found in the “Computer Code S2” document of the Appendix A. 

After applying PCA, the jackknifing step revealed that the Rep value that generated the most robust model (98.1% of the *jackknifing* models correctly classified all the *jackknifing* samples) was 20, and the model contained 20 variables. Therefore, 20 features were chosen as relevant to the classification; the *m*/*z*-values of those features were: 150.0, 152.0, 175.0, 193.0, 203.2, 205.1, 215.0, 216.0, 237.0, 345.2, 385.2, 399.2, 413.1, 430.2, 431.2, 444.2, 445.2, 859.3 and 971.2 *m*/*z*.

The first validation step revealed that the number of PCs that could be used to correctly assign the first validation subset was every number in between 3 and 20. However, if the number of PCs is the same as the number of variables, there is a high chance of overfitting. Thus, according to the workflow explained in “Appendix A—First Validation of the *k*-means clustering model” in Appendix A, the number of PCs chosen was 10, as adding more PCs would lead to overfitting of the data.

For the second validation, the data matrix was multiplied by the loadings and then classified by the SVMs.

### 2.5. Results of PCA of the Small Molecules Found in Snake Venom

#### 2.5.1. *k*-Means Clustering

The results of the analysis performed on this model can be found in “Appendix A—Results of the *k*-means clustering model” in Appendix A.

Although the optimized model was able to correctly classify most of the samples, only 50% were classified as hard hits. This lack of certainty could be explained by the overall trend of the Viperidae family not being defined by a single direction in any of the PCs—in contrast to what we found for Elapidae. This distribution over two PCs can result in lower classification power, as more extreme values for the seven features—the ones defined as relevant by the model—within the Viperidae family lead to a scattering of the samples all over the subspace created by the PCs (instead of close clustering), rather than an increase in density of the Viperidae cluster.

#### 2.5.2. Support Vector Machines

As mentioned, a classifier based on SVMs with Rep = 20 and n° PCs = 10 was built. A 2D representation of this model, including the scores projected onto the two first PCs, can be found in Figure 2. Two PCs were chosen for this representation, as it is easier to visualize, but the actual subspace is contained in 10 dimensions, as they were the ones chosen to describe most of the variance of the model.

A table containing the values for all the loading values is included in “Appendix A: Relevance of each feature” in the Appendix A.

The clustering in Figure 2 already looks more organized than the one we could see for the model obtained when Rep was set to 36 (Appendix A of “Appendix A—Results of the *k*-means clustering model” in Appendix A, where the representation of the optimized *k*-means model can be found). Whereas Elapidae scores seem to distribute most of their variance in one direction—almost solely along the second PC—Viperidae scores are distributed along both PCs, although mostly along the first one. Figure 2 also shows how non-spitting cobras seem to have high values for PC1 and low values for PC2 compared to the rest of the Elapidae family, which indicates that further classification could probably be performed when a bigger dataset is used.

To understand the information held in the PCs, their loadings are represented in “Appendix A—Representation and analysis of the Loadings in the SVM model” in Appendix A. The first principal component gives similar relevance to each of the features, using all of them to define most of the variance in the model. However, while features 1, 2 and 20 (the ones with weights higher than 0) are the only ones to be considered when classifying a sample as a member of the Elapidae family, features 6, 7 and 8 (the ones with the lowest weights) are essential when classifying a sample as Viperidae and explaining most of their variance.

It is interesting to look at all the features and samples at the same time to better understand the patterns within the data; therefore, the autoscaled values of those 20 features were summarized in a heatmap depicted in Figure 3.

Similar to what we found when Rep = 36 was used to create the model (Appendix A of “Appendix A—Results of the *k*-means clustering model” in Appendix A), the first features and the last one (150.0, 152.0, 175.0, 193.0 and 971.2 *m*/*z*) seem to be consistently more abundant in Elapidae than in Viperidae. Because the last feature (971.2 *m*/*z*) also holds some extreme variability within the Elapidae family, but its intensity can be similar to that of Viperidae in some cases, this feature is pushed back to the fifth component. Nonetheless, the first four variables—with quite different values between the two families—are found in the second principal component and thus explain much more variance. As can also be seen in the heatmap, features 7 and 9 seem to have somewhat of an influence when defining the Elapidae. These variables are held inside the second, third and fifth components. If we project the scores of those second and third PCs—only the most important two out of the three are used for clarity in the representation—the resulting graph (Figure 4) should explain most of the variance found within Elapidae. Only the said family is represented for clarity, as introducing the Crotalus family would make the image too crowded, and it would add no additional value.

It is noticeable to mention that the feature with 215.0 *m*/*z* seems to be able to clearly differentiate between *Dendroaspis* and *Naja* while also differentiating *Dendroaspis* from *Crotalus*, as occurred in the *k*-means model. However, no PC relies on this variable to generate such clustering, probably due to the low number of *Dendroaspis* samples, which biases the model generation. PC 3 does take this variable into account, but, because this principal component also considers the feature with 237.0 *m*/*z*, the ability of the model to cluster *Dendroaspis* separately is strongly diminished, as can be seen in Figure 4.

Most of the variability found in Elapidae useful for classification can be essentially summarized in features 1, 2, 3, 4, 7, 8 and 9, which—by looking at Figure 4—means that these two PCs should be sufficient for explaining most of the relevant variability. However, the feature with 215.0 *m*/*z* is not relevant enough in these PCs for us to see the clustering of *Dendroaspis*, probably due to the aforementioned bias. However, by adding more samples to the model, this variable should stand out as a paramount feature to define and cluster the said group. As will be described in the Section 3, this feature is thought to derive from the Deoxyribose 5-monophosphate metabolite, commonly known for partaking in the pentose phosphate pathway, which is used to anabolize precursors for the synthesis of nucleotides and to maintain the redox state of the cell [25].

The PCs in the model that hold the highest variance and that contain information concerning the features that define the variability found within Viperidae are the first PC (as it holds information concerning variables 5 to 19 and explains most of the variability within the model) and the second one (as it contains information concerning the variability of features 3 and 4, which also define Viperidae). These PCs can be found in Appendix A of “Appendix A—Representation and analysis of the Loadings in the SVM model” in Appendix A.

After analyzing the model built for Rep = 20 and n^o^ PCs = 10 (Figure 2), the samples within the second validation subset were projected onto this model, thereby obtaining their scores. In Figure 5, the scores of both validation subsets were included with those already in the model. Only snakes from the Viperidae family were chosen for the second validation model to test its ability to predict the family of the snake from which a specific venom originates. It is relevant to mention that some of the said samples derive from a genus that was not included in the model—*Bothrops*—which was done to test the ability of the model to predict the family of a genus, which had not been presented to the model. In brief, this means that if the model was able to correctly classify those samples, the features found—and their concentrations—could be assumed to be homogeneous throughout the different families. However, this statement should be explored in further studies, as the prediction of one genus, which was not included in the model, cannot be extrapolated to all other genera.

The results of the validation demonstrated that 100% of the second validation subset samples were correctly classified, thus showing that the SVM model rendered excellent results, whereas the *k*-means approach did not. Even samples coming from a genus unseen by the SVM model (*Bothrops*) were correctly identified within the Viperidae family. As can be seen, the representation of the second validation shows how the clustering defined by the hyperplane works, as the distribution of the validation samples does not seem to follow a random pattern, but rather, the same trend, which could be seen in the model itself, can be found for the validation subsets.

To understand the reason for this clustering, the features present in the model were looked into more thoroughly in order to recognize the metabolites that were associated with the said features, and thus, the metabolites that were able to discriminate between different taxa.

In the first place, the identification of the metabolites was performed via comparison of the *m*/*z*-values of the features with two databases of metabolites; one included the ones found in snake venom and the other one metabolites only found in humans [8,26]. As a second step, the metabolites from which we had stock solutions were spiked onto the venom samples, which contained features to check whether their intensity increased after the addition. Because we did not have stock from all expected metabolites, LC–MS/MS was performed on a pooled sample containing four of the venoms (*Naja Nigricinta*, *Crotalus vegrandis (2)*, *Crotalus culminatus* and *Crotalus lorenzoensis*), which, in conjunction, contained all the metabolites relevant to the model. With the acquired data, the MS/MS spectra resulting from fragmentation of accurate masses that correlated to features of the specific metabolites under investigation, could be compared with MS/MS fragmentation spectra retrieved from the Human Metabolome Database (HMDB) [26], which contains all human metabolites found and relevant information about them, such as their mass and fragmentation spectra. This database was chosen, as it is the only one that includes MS/MS fragmentation spectra of a high number of metabolites (220,945). When a match was found for a feature’s accurate mass and its MS/MS spectrum, that feature was designated as the metabolite listed in the database.

In Table 2, the names, retention times and *m*/*z*-values of each metabolite relevant to the model can be found. The second-to-last column accounts for the confidence level at which the metabolites were determined, and the last column specifies the genera for which that feature was found in more than 50% of the genera samples. The confidence level was assigned by means of standards within the Metabolomic Standards Initiative (MSI) [27]. Level 1 confidence corresponds to validated identification—confirmation by use of a standard. Level 2 confidence signifies that both the *m*/*z*-value and the MS/MS pattern correspond to those of the mentioned metabolite. The level of confidence “- “ corresponds to lack of confidence level assignment, as only *m*/*z*-values were cross-referenced. Because Villar-Briones and Aird [8] (referenced as (1) in Table 2) analyzed metabolites in venoms, whereas Wishart et al. (HMDB) [26] (referenced as (2) in Table 2) took into account metabolites only found in humans, it is notable to mention that the metabolites in accordance with Villar Briones and Aird’s data are expected to be identified more reliably due to the fact that we are dealing with snake venom metabolites and not human metabolites.

All relevant metabolites for the model eluted from LC within the first 20 min of the LC–MS analyses, which means that the analysis time could probably be adapted and shortened after this time threshold without losing any crucial information when the analyses are performed for snake venom family classification by means of the defined SVM model.

More confidence in the identification of metabolites with no MSI confidence level assignment would be needed to robustly elucidate the specific venom components that underpin the differences between different snake genera.

For further understanding of the patterns found in snakes’ venom metabolites, the relevant features of all analyzed samples were explored in the heatmap found in Figure 6.

## 3. Discussion

Several patterns could be found regarding the intensities of these features, which the model found relevant. These patterns were able to differentiate between families and even genera and clades, meaning that there is probably a biological reason as to why the scores of these samples are distributed in such manner. It is important to mention here how the model is able to group both *Bothrops* and *Crotalus* venoms under the same family—Viperidae—even though the intensity of several features does not comply with the general trend of Viperidae.

When analyzing the patterns found in the intensity of each feature, the fact that citrate was mostly seen in Viperidae venoms, whereas it was not found in *Dendroaspis*, stands out. Citrate is known for its protease inhibition activity, due to which it is thought to be present in snake venom to prevent degradation of the venom by its own metalloproteases [28]. Additionally, this study also reported that citrate was able to inhibit a PLA_2_, which was investigated. As can be seen in Figure 6, citric acid is present, on average, in higher concentrations in the Viperidae family, which is known to have venom compositions that contain, on average, much higher concentrations of proteases than Elapidae venoms [29]. This is in accordance with other studies, which analyzed citrate in snake venom [8,28]. This metabolite is known to work by chelating metals, such as zinc, thus being able to inactivate metalloproteases residing in the venom gland. Additionally, it lowers the pH of the matrix, thereby lowering the enzymatic activity of metalloproteases. Once a venom enters the prey’s body, citrate can possibly also assist in disrupting the normal coagulation processes by chelating Ca^2+^, the most important metal ion in the coagulation cascade [30]. Citrate is therefore suggested to be present in metalloprotease-rich venoms to prevent self-harm while potentially aiding in their hemotoxicity once the venom enters the prey’s bloodstream. Spitting cobras can use their venom as a defense mechanism, thus, their venom aims toward cytotoxicity [31]. These venoms have low amounts of metalloproteases [32], and, with the hypothesis postulated above, it makes sense that almost no citrate was found in these venoms. Endogenous peptides, such as pENW, are also known to be able to inhibit metalloproteases [12,33,34,35], which could be a reason as to why they were mostly found in hemotoxic Viperidae venoms. The presence of these glutamate-containing oligopeptides in the venom is known to contribute to reducing proteolytic activity of the SVMPs, thus preventing venom degradation by its own toxins [35], which could explain the correlation between the concentration of these oligopeptides and SVMPs found in Viperidae venom.

*Naja* venoms—especially those from African non-spitting cobras—on average contained higher methionine concentrations than the other taxa. This metabolite is not found in the rest of the genera, meaning that methionine is not a trait of the Elapidae venoms but rather specifically of the *Naja* venoms.

*Dendroaspis* is the genus, which contained the least amount of metabolite with a feature that had a *m*/*z*-value of 215.02. As stated, this feature is suspected to derive from deoxyribose 5-monophosphate, which is known to take part in the pentose phosphate pathway (PPP). PPP is a metabolic pathway involved in the generation of NADPH and pentoses, and it has been demonstrated as a major regulator of cellular redox homeostasis and nucleotides biosynthesis [36].

Thirteen features and metabolites were found in most of the *Crotalus* genera, and metabolite identification of those features—with MS/MS and/or standards—would be needed to draw further conclusions. Tryptophan, an uncommon metabolite in other taxa, was frequently observed in the *Crotalus* venoms, for which we currently do not have a hypothesis.

There does not seem to be any specific metabolite able to fully define only the *Bothrops* genus. This might be due to the fact that only three *Bothrops* samples were analyzed and that these samples were not used in the model. However, even if the defined SVM model was not able to indicate which metabolites are unique for the said genera, it was indeed able to classify them as Viperidae due to the features these samples contained and their respective intensities.

Considering classification regarding the origin of the metabolites previously stated in the Introduction (endogenous metabolites produced by the venom gland, endogenous metabolites unrelated to venom and exogenous metabolites), it is interesting to note that endogenous metabolites, which could be related to the snake’s metabolism rather than involvement in venom functioning—such as methionine and tryptophan—had different abundance levels between Elapidae and Viperidae. The only endogenous metabolite identified probably related to venom activity and/or venom gland protection (citric acid) followed the expected pattern of taxa differentiation based on the biological relevance of the said analyte within the venom. No exogenous metabolites were identified in this study within the set of features in the model.

In addition to creating a model able to discriminate between Elapidae and Viperidae based on their metabolites, this research can also be considered as an untargeted metabolomic study, as a list with all the features found (i.e., peaks detected in MS and their corresponding *m*/*z*-values and intensities) in all the venoms measured was created and included in “Appendix A: Level of each metabolite” in the Appendix A. These types of studies are helpful when identifying core metabolites in the venoms but also for elucidating the theories explaining why those species are biologically relevant. This study has certain limitations that allow for bias to potentially modify the results. For example, factors such as sex, age and/or geographical location were not considered in the model building in this study, as these data were not available for a large part of the venoms in our venom collection, and/or some venom samples were derived from pooled milkings of different individuals. Additionally, some of the analytes were found at the beginning of the chromatographic analyses—eluting almost with dead volume—which is known to be detrimental for their ionization in the MS due to the high number of species being ionized during this time frame. Because no traceback of those factors—and many others—could be performed, the samples must be assumed to represent the totality of their family, genus, species and subspecies. However, this study is the first attempt to initiate research that focuses on the metabolites found in venom, as they can also be useful for understanding these complex mixtures.

In order to be able to investigate specific metabolites and/or features, a dashboard tool can be used to easily visualize the variables of choice. For this, “Appendix A: Level of each metabolite” in the Appendix A is used and imported into the dashboard tool. We performed this for our dataset, and all the relevant representations of the data can be found in https://public.tableau.com/app/profile/luisce99/viz/Metabolome-basedclassificationofsnakevenomsbybioinformatictools/Dashboard1, (accessed on 11 February 2023). Due to the intrinsically interactive properties of the dashboard, the information selected can be sorted and presented in different representative ways, such as bar graphs and pie charts. This implies that users can choose the metabolites and/or features they want to investigate and from there be able to immediately obtain information regarding their abundance level in each venom, or in a clade or taxa. Examples of visualizations presented using the dashboard can be found in “Appendix A—Examples of data visualization with the dashboard” in Appendix A.

This software is able to automatically sort the input data into different variables, clean and interpret the data, and understand SQL-type queries and implement them into one’s dataset. It can also sort the data in different tables and create new variables with unions or intersects between different tables. The main reason this software was also used in this study is due to its ability to easily create quick representations of the input data and merge them into a single dashboard with all the required information represented in clearly structured overviews.

## 4. Conclusions

A high-throughput analytical LC–MS-based pipeline for analysis and automated extraction of all masses corresponding to small molecules in the range of 100 to 1000 *m*/*z*-values in snake venoms was set. Within this workflow, a comprehensive list containing all the features found and their characteristics—*m*/*z*-values, retention times and intensities—in the samples was obtained. This list can be found in “Appendix A: Level of each metabolite” in the Appendix A.

After investigating two ways of classifying snake venoms by their metabolites (*k*-means and SVMs), the SVM approach showed not only much more robustness but also much more reliability when attempting to determine whether a venom originated from an Elapidae or a Viperidae snake. This is probably because, as we transform the sample matrix into a score matrix, the two groups are naturally separated within the subspace of the loadings. Thus, the SVM—being a binary linear method—recognizes this separation and takes advantage of it.

From the results of the validation of the model, it can be inferred that metabolites are consistently present in the snake venoms and that their levels are sufficiently similar between samples from the same family (and sufficiently different between families) to be used for family classification of snake venoms. This differentiation can also be found—with less confidence—within other taxa (genera, clades, etc.). MS/MS analysis of the said metabolites would lead to improvement in terms of their identification level. Some of the metabolites that could be identified presented distinctly different concentration levels in venoms of different families due to potentially explainable biological relevance. Hypotheses regarding the said levels are proposed in this study.

To further develop the project, analyses of additional and different venoms would be needed. By analyzing more venoms, the classification could reach and recognize deeper patterns in the data, thus theoretically being able to also fully classify venoms into subfamily, genus and eventually maybe even the diet of a specific snake. In order to achieve this, the said venoms should also include biological and geographical information, such as age, gender, location, etc., from the snake they came from, so that the result is more representative. Follow-up targeted studies could also be performed on the metabolites identified, as understanding the biological reasons for the presence of some of these small molecules in venoms could provide insightful information concerning snake venom.

This study mainly focused on setting up and demonstrating the analytical methodology to obtain a coherent list of metabolites and the venoms they are found in by means of developing a family classifier. Future studies should focus on the metabolite variation related to morphological and ecological parameters.

## 5. Materials and Methods

### 5.1. Chemical and Biological Reagents

Water was purified with a Milli-Q Plus system (Millipore, Amsterdam, The Netherlands). DMSO was supplied by Riedel-de-Haen (Zwijndrecht, The Netherlands). Acetonitrile (ACN; ULC/MS grade), trifluoroacetic acid (TFA) and formic acid (FA) were obtained from Biosolve (Valkenswaard, The Netherlands). All salts used for buffer preparation were of analytical grade and bought from Merck (Kenilworth, USA), Fluka (Bucharest, Romania) or Sigma-Aldrich (Darmstadt, Germany). Micro-90^®^ concentrated cleaning solution was supplied by Sigma-Aldrich. Lyophilized snake venoms (Worksheet 1 of Workbook 1 of SI) were provided by the Centre for Snakebite Research & Interventions (Liverpool School of Tropical Medicine (LSTM), UK) and the historical collections of Freek J. Vonk (FV) and Manjunatha Kini (K) and stored on a long-term basis at −80 °C. Stock solutions of crude venoms (5.0 mg/mL) were prepared in water prior to analysis and stored at −80 °C. A total of 50 venoms were studied; 15 of those venoms originated from the genus *Naja* and 5 from the genus *Dendroaspis*, both within the Elapidae family. A total of 27 of those venoms originated from the genus *Crotalus* and 3 from the genus *Bothrops*, both within the Viperidae family. Volumes of 50 μL of 1 mg/mL solutions of each venom were injected into the HPLC-MS. Solutions of Phenylalanine, Caffeine, d9-Caffeine, Nortriptyline and Metoprolol (10 mM) were prepared in Milli-Q (except for Nortriptyline, which was dissolved in DMSO—Riedel-de Haen) and stored at −80 °C. They were used as internal standards for this study due to their small size, the ability to absorb at the wavelengths studied and disparate retention times. All these chemicals were bought from Sigma Aldrich.

The origin of the lyophilized venoms, the order in which they were injected and the model the samples were used on are included in “Appendix A: Order of the injections” in the Appendix A.

### 5.2. Analysis of Small Molecules

#### 5.2.1. Separation

Reverse-phase high-performance liquid chromatography (RP-HPLC, or LC for short) was carried out using a Shimadzu HPLC system managed by Shimadzu LabSolution software (Shimadzu, s-Hertogenbosch, The Netherlands). Four types of samples were analyzed in the order specified in Table 1 of the SI: 50 μL water injections acting as blanks, 50 μL of each of the mentioned venoms spiked with d9Caffeine (2.5 µM; used as a standard), a mixture of standards (Phenylalanine 2.5 μM, Caffeine 250 nM, d9-Caffeine 250 nM, Nortriptyline 250 nM and Metoprolol 250 nM) and 50 μL of Naja siamensis venom spiked with the said mixture to test the repeatability of the method and the capability of those standards to normalize the intensities of the features found in *Naja siamensis*. All samples were injected using a SIL-30AC autosampler, and the column used was a Waters Xbridge Peptide BEH300 C18 analytical column (100 × 4.6 mm, 3.5 µm particle size and 300Å pore size with a flow rate of 0.5 mL/min). Mobile phase A consisted of 97.9% H_2_O, 2% ACN, 0.1% FA, and mobile phase B consisted of 97.9% ACN, 2% H_2_O, 0.1% FA. The gradient program was set as follows: linear increase to 50% B in 30 min, followed by linear increase to 90% B in 4 min, isocratic elution for 5 min at 90% B, down to 5% B in 1 min and then equilibration for 10 min.

#### 5.2.2. Detection

For all LC–MS analyses, a flow splitter was added after separation. 90% of the flow was directed toward an SPD-20A Prominence UV Detector to record UV spectra at both 220 and 254 nm. The remaining 10% of the flow was directed toward a MaXis QTOF mass spectrometer (Bruker Daltonics, Bremen, Germany) hyphenated to the HPLC via an ESI source operating in positive ion mode. The following parameters were used: temperature 220°C, capillary voltage 4.5 kV, gas flow 8.0 L/min, Nebulizer pressure 1.8 Bar. The spectra were stored at a rate of 2 Hz in the range of 100 to 1500 *m*/*z*-values. otofControl software was used for instrument control (Version 5.2-Build 0.9; Bruker, MA, US).

To acquire MS/MS data on the relevant metabolites found in the model, an Impact II mass spectrometer (Bruker Daltonics, Bremen, Germany) in data-dependent mode was hyphenated to the HPLC via an ESI source operating in positive ion mode. The following parameters were used: temperature 380 °C, capillary voltage 4.5 kV, gas flow 8 L/min, Nebulizer pressure 1.8 Bar, CI gradient from 25 to 55 kV.

#### 5.2.3. Data Processing

Data processing consisted of several steps. First, the output data from the MS were converted to mzXML. This was performed by the MSConvert software with the thresholds specified in “Appendix A—Thresholds applied on MSConvert” in Appendix A. Once the mzXML files were obtained, both SAFD—an algorithm able to recognize features by fitting a 3D Gaussian able to align the features within one sample by means of their *m*/*z*-values, retention times and intensities [37]—and the algorithm given in “Computer Code S1” in the Appendix A were used to identify the features in each sample and align them into a coherent matrix. The latter based the said alignment on similarities between retention time windows and *m*/*z*-values. With this information, it was possible to recognize the same feature in different venoms, thus being able to align the same feature in all venoms, thus being able to create the mentioned coherent matrix with all venoms and features. After the coherent matrix was developed, the measured samples were divided in three subsets in order to perform cross-validation of the model. After the validity of the model was proven, the model was looked at to check what features were recognized as relevant, and their concentrations in each venom were then investigated. This step was performed in conjunction with the identification of several metabolites associated with those features relevant to the model by means of the acquired MS/MS spectra of these metabolites or by standard addition when a metabolite was commercially available.

## Data Availability

The data presented in this study are available in this article and Appendix A.

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
