# Peer review of "Metabolome-Based Classification of Snake Venoms by Bioinformatic Tools"

_toxins, 2023, doi:10.3390/toxins15020161_

Round 1
Author Response
Reviewer 1 – Comment 1) Please remove the italics from the words ‘Elapidae’ and ‘Viperidae’, keeping only in the genera, species, and subspecies.
ANSWER: We thank the reviewer for these well-observed details. They have been addressed.
Reviewer 1 – Comment 2) Page 3, line 3: Bold "figure 1" and unbold the beginning of the word 'order'. 2
ANSWER: We thank the reviewer for these well-observed details. They have been addressed.
Reviewer 1 – Comment 3) Page 2, lines 64 – 66: In my opinion, more than classifying venoms of different taxa, this type of study has the potential to help understand the importance of the metabolites present in snake venoms for the human envenoming and, eventually, for the ecological relationships between venomous snakes and its prey/predators. Obviously, important information such as differences related to sex, age, and geographical distribution should be considered in the analyzes.
ANSWER: We thank the reviewer for this comment and addressed it by making the following changes: “One of the purposes of this research will be looking at classification of small molecules in venoms depending on the source of said metabolites, which can be useful for the classification of snake venom” was turned into “One of the purposes of this research will be looking at classification of small molecules in venoms depending on the source of said metabolites, which can be useful for the classification of snake venoms, but also for understanding the relevance of small molecules in this biological matrix by considering how they might be involved in for example human envenoming, maintaining venom gland integrity, and ecological relationships between snake and prey.”.
As this study was for a large part focused at setting up and demonstrating the methodology, we used venom samples available to us from our in-house snake venom library. Many of the venoms in our library do not have information on age, sex and geographical distribution. Future studies are in preparation where this will be taken into account. We added the following sentence to the conclusions (line 597-600): “This study mainly focused at setting up and demonstrating the analytical methodology used to obtain a coherent list containing the metabolites found in the studied venoms by means of developing a model able to classify snake venoms at the family level. Future studies should focus on the metabolite variation related to morphological and ecological parameters.”
Reviewer 1 – Comment 4) Page 7, lines 294 – 299: Perhaps bit different finds could be obtained for Viperidae family in term of variance, if enough bothropic venoms had been included in the analyses, to build the model, and not only in the validation step, as it was done. In addition, the inclusion of a reasonable number of poison samples from different species of Bothrops would certainly make the work more comprehensive and representative of the Viperidae diversity.
ANSWER: We thank the reviewer for this comment. Indeed, we agree with this. However, as mentioned in our reply to the previous comment, the used venoms were available to us from our in-house library. We only had access to several Bothrops venoms. The focus of this study was to optimize and demonstrate the methodology. We believe we succeeded in this. In addition, the fact that Bothrops was recognized as Viperidae without being included in the model is an interesting outcome, meaning that the model -apparently- was able to indeed identify the key “Viperidae” small molecules.
Reviewer 1 – Comment 5) Page 8, line 315: In the legend of the figure 3, the authors should include additional information allows the reader to identify the "subspecies” or replace this term for ‘subclade’ as shown in the figure 1.
ANSWER: We thank the reviewer for pointing to this detail. It was indeed an erratum. It has now been replaced by “subclades”.
Reviewer 1 – Comment 6) Page 14, lines 475 – 477: the authors should include in Table S.2 (supplementary material) the full scientific names of the Crotalus subspecies whose venoms were used in the analyses. In addition, information about the types of venoms (type I - hemorrhagic, type II - neurotoxic or intermediate, such as Crotalus simus venom) of these subspecies should also be considered in data analyses and discussion since they can differ markedly in both, protein composition and lethality.
ANSWER: We thank the reviewer for coming up with an idea that could add extra information to the research. However, we don’t fully agree with adding it to the current manuscript. It would be interesting indeed, but it might also be confusing for the reader to talk only about phylogenetic differentiation of the venoms throughout the manuscript, and then suddenly add some other factors that might also differentiate specifically the Crotalus family. Also, the information pointed out by the reviewer is information that is not well documented for all studied venoms, and many venoms have combined pathologies -making the differentiation complex.
Regarding the subspecies, we only mentioned them where this was either known for our venom collection, or relevant. Otherwise, we only mentioned the species level.
Reviewer 1 – Comment 7) Page 14, line 509: Please join the comma and parenthesis.
ANSWER: We thank the reviewer for pointing out this typo. It has been corrected.
Reviewer 2 Report
The research topic is quite interesting and relevant but the manuscript needs some editing. For example:
Line 29-30: This bit is an extraneous comparison more directed towards creating a dramatic effect. The facts and figures presented represent the significance of snakebites in the tropics.
Lines 30-35: They are mostly redundant. Rephrasing may help.
The reasoning of why Crotalus and Bothrops were selected within Viperidae and Dendroaspis and Naja is missing. The authors should give explain why they chose only these genera were selected. Similarly, for the selection of African clade within Naja.
Line 92-104: They are extraneous and do not add anything to the introduction. If absolutely needed they can be made more specific to be added to the methods section but right now lines 97-104 are very abstract.
Lines 105-117: Need to be rephrased to be more specific. Currently, it’s quite tricky to gather the clear objective and goals of this study.
Line 106: When using any abbreviation, it’s best practice to first give its unabbreviated name.
Line 119: Needs to be rephrased
Line 120: What were the “found features” please be specific.
Lines 126-131: belong in the methods section and need to add more details.
Line 134: Did the authors check for another species let's say a Viperidae? Why did the author(s) select Naja Siamensis? There needs to be an explanation.
The results and methods section are interchangeably used. This needs to be rectified.
The methods section is seriously missing literature citations. The methods section needs to be rewritten as many necessary details are missing or phrasing is not properly done.
Line 600-602: The venom coming from Crotalus vs Bothrops genera is quite disparate, similarly for Naja vs Dendroaspis. Please explain. Currently, the study design seems biased.
Also, specify which species the venom came from. What was their age, etc.? Venom varies with age, habitat, sex, and diet within species.
Line 606: Explain internal standards.
Line 602-606: Needs citation
Line 606-607: Not needed.
Line 621: Why did the author(s) select Naja Siamensis? There needs to be an explanation.
Line 630: Please delete the main part of this study.
Section 5.2.3: Is missing necessary details but has extraneous details. Please rectify.
Author Response
Reviewer 2 – Comment 1) Line 29-30: This bit is an extraneous comparison more directed towards creating a dramatic effect. The facts and figures presented represent the significance of snakebites in the tropics.
ANSWER: We thank the reviewer for this comment, this fragment of the manuscript has been deleted.
Reviewer 2 – Comment 2) Lines 30-35: They are mostly redundant. Rephrasing may help.
ANSWER: We thank the reviewer for this comment and now address it as follow: “Although snakebite envenoming can be found in all inhabited continents, it is mostly a public health concern in tropical and sub-tropical areas, and in part this is due to the lack of health system resources found within developing countries. One of the factors involved in the mortality of snakebite in these countries comes from the lack of availability -or affordability- of high quality specific antivenoms, which is caused by the complexity of snake venoms.” was turned into “Although snakebite envenoming can be found in all inhabited continents, it is mostly a public health concern in tropical and sub-tropical areas in developing countries, and in part this is due to the lack of health system resources found within these countries.” in lines 29-32.
Reviewer 2 – Comment 3) The reasoning of why Crotalus and Bothrops were selected within Viperidae and Dendroaspis and Naja is missing. The authors should give explain why they chose only these genera were selected. Similarly, for the selection of African clade within Naja.
ANSWER: We appreciate the comment of the reviewer. One of the reasons for selecting the venoms that were included in our study was simply based on availability. Our venom library includes an extensive number of Crotalus, Dendroaspis and Naja venoms, and several Bothrops venoms. From other Viperidae and Elapidae genera, we do not have many venom samples in our venom library. However, we feel this is not an issue for the current study as it was aimed at setting up a workflow to work with and understand variability in small molecules within venoms. Currently, we are setting up future studies with other groups who have access to large numbers of venoms with associated biological, geographical, age and habitat data.
Reviewer 2 – Comment 4) Line 92-104: They are extraneous and do not add anything to the introduction. If absolutely needed they can be made more specific to be added to the methods section but right now lines 97-104 are very abstract.
ANSWER: These lines do indeed give some extraneous information, which were already included in the manuscript throughout the results section. We now deleted the previous lines 96-101 from the manuscript to reduce the confusion these lines can cause in the reader when added in the introduction.
Reviewer 2 – Comment 5) Lines 105-117: Need to be rephrased to be more specific. Currently, it’s quite tricky to gather the clear objective and goals of this study.
ANSWER: We thank the reviewer for this comment and now changed in lines 105-111: “The automated extraction of data from the LC-MS analyses, which is always performed when following the analytical workflow, also results in the creation of a comprehensive list containing all m/z-values and retention times of all compounds detected in each venom within the used mass range for detection (i.e. 100 to 1000 in our case). Thus, each row contains all of the features contained in each sample.” into “The automated extraction of data from the LC-MS analyses, which is always performed when following the analytical workflow, also results in the creation of a comprehensive list containing all m/z-values and retention times of all compounds detected in each venom within the used mass range for detection (i.e. 100 to 1000 in our case). Thus, by following the analytical workflow, two main results are obtained: a classifier able to predict the family of the snake (in this case between Elapidae and Viperidae), and a comprehensive list that contains the processed analytical data on all small molecules found for each venom.”
Reviewer 2 – Comment 6) Line 106: When using any abbreviation, it’s best practice to first give its unabbreviated name.
ANSWER: We closely inspected our complete manuscript for this and indeed see that we missed the unabbreviated name for LC-MS when first mentioned. This has now been corrected. The following correction was made: “LC-MS“ is now written as “Liquid Chromatography – Mass Spectrometry” the first time it is introduced.
Reviewer 2 – Comment 7) Line 119: Needs to be rephrased
ANSWER: We thank the reviewer for this comment and now address the apparently unclear sentence as follows: “The overall pipeline started with the LC-MS analysis of the small molecules in 50 different venoms in duplicates, followed by extraction and alignment of the found features (i.e., all detected peaks in MS and their corresponding properties such as m/z-value and retention time) for all the analyses.” was changed into “The overall workflow started with the LC-MS analysis of the small molecules in 50 different venoms -in duplicate, followed by extraction and alignment of the found features (i.e., all detected peaks in MS and their corresponding properties such as m/z-value and retention time) for all the analyses.”
Reviewer 2 – Comment 8) Line 120: What were the “found features” please be specific.
ANSWER: We think the reviewer might have missed the sentence where this is explained, which can be found after the concept is introduced for the first time in lines 118-121. It indicates the following: […] followed by extraction and alignment of the found features (i.e., all detected peaks in MS and their corresponding properties such as m/z-value and retention time) for all the analyses.“
Reviewer 2 – Comment 9) Lines 126-131: belong in the methods section and need to add more details.
ANSWER: This study focuses on developing a new analytical method. Therefore, some of the described texts in the results and/or methods section can look like they are in the wrong section. What we did to overcome this as much as possible is to include in results section everything that we did not optimize/modify. Such as the MS/MS parameters, which are included in the Methods section.
However, since the Methods section was apparently not fully clear to the reviewer, the following text has been modified from: “Once the mzXML files were obtained, both SAFD -an algorithm able to recognize features by fitting a 3D Gaussian able to align the features within one sample by means of their m/z-values, retention times and intensities [38]- and the algorithm given in Computer Code S.1 in the Supporting Information-, were used to identify the features in each sample and align them into a coherent matrix.” into: “Once the mzXML files were obtained, both SAFD -an algorithm able to recognize features by fitting a 3D Gaussian able to align the features within one sample by means of their m/z-values, retention times and intensities [37]- and the algorithm given in Computer Code S.1 in the Supporting Information-, were used to identify the features in each sample and align them into a coherent matrix. The latter bases said alignment in similarities between retention time windows and m/z-values. With this information, it is possible to recognize the same feature in different venoms, thus being able to align the same feature in all venoms, thus being able to create the mentioned coherent matrix with all venoms and features. After the coherent matrix is developed, the measured samples were divided in three subsets in order to perform cross-validation of the model. After the validity of the model was proven, the model was looked at to check what features were recognized as relevant, and their concentrations in each venom were then investigated.”
Reviewer 2 – Comment 10) Line 134: Did the authors check for another species let's say a Viperidae? Why did the author(s) select Naja Siamensis? There needs to be an explanation.
ANSWER: We thank the reviewer for this comment. An explanation has been added to the manuscript: in the line 140-141: “This venom was chosen arbitrarily due to the high stock we had, as the same function could be attained by any other venom.”
Reviewer 2 – Comment 11) The results and methods section are interchangeably used. This needs to be rectified.
ANSWER: This has already been addressed in our ANSWER to comment 9 of the same reviewer.
Reviewer 2 – Comment 12) The methods section is seriously missing literature citations. The methods section needs to be rewritten as many necessary details are missing or phrasing is not properly done.
ANSWER: We find it difficult to address this comment. The comment is not giving any specification at all on what parts or sentences in the Methods section need to be rewritten, why they need to be rewritten, and which necessary details are missing. We feel that the methods section is adequate and describes all needed information. Additional information on the analytical methodology presented is given in the Supporting Information to prevent making the text too lengthy.
The other two reviewers explicitly mention that the study is clearly written and presented, and that the methods section is adequate.
We carefully went through the Methods section to search for missing details and/or unclearly formulated sentences. We now extended the last Methods section paragraph in the following way in lines 650-676: “Data processing consisted of several steps. First, the output data from MS was converted to mzXML. This was done by the MSConvert software with the thresholds specified in Section 6 – Thresholds applied on MSConvert Document S.1 of the Supporting Information. Once the mzXML files were obtained, both SAFD -an algorithm able to recognize features by fitting a 3D Gaussian able to align the features within one sample by means of their m/z-values, retention times and intensities [38]- and the algorithm found in Computer Code S.1 in the Supporting Information-, were used to identify the features in each sample and align them into a coherent matrix.” was changed into “Data processing consisted of several steps. First, the output data from the MS was converted to mzXML. This was done by the MSConvert software with the thresholds specified in Section 6 – Thresholds applied on MSConvert – of Document S.1 of the Supporting Information. Once the mzXML files were obtained, both SAFD -an algorithm able to recognize features by fitting a 3D Gaussian able to align the features within one sample by means of their m/z-values, retention times and intensities [37]- and the algorithm given in Computer Code S.1 in the Supporting Information-, were used to identify the features in each sample and align them into a coherent matrix. The latter bases said alignment in similarities between retention time windows and m/z-values. With this information, it is possible to recognize the same feature in different venoms, thus being able to align the same feature in all venoms, thus being able to create the mentioned coherent matrix with all venoms and features.”
Reviewer 2 – Comment 13) Line 600-602: The venom coming from Crotalus vs Bothrops genera is quite disparate, similarly for Naja vs Dendroaspis. Please explain. Currently, the study design seems biased.
ANSWER: We appreciate the concern of the reviewer. However, as mentioned also as reply to previous comments (such as Comments 9 and 11 of this reviewer and Comment 3 from Reviewer 1), the focus of this study was at developing a new analytical method (with venoms available to us). Only a maybe almost non-biased study can be performed if for all venomous snake genera many venoms were included in the study. This in our opinion is practically impossible to perform which is related to acquiring all these venoms (availability and cost issues). Additionally, issues rise on other aspects such as the Nagoya protocol. We remove as much bias as possible (with the dataset available to us) by performing multiple validations with datasets not included in the building of the model. Even though the venoms are different between each other, it would seem like they are more similar within the same family (even though they come from different genera), than they are similar within the same order. In other words, it would make sense for a Dendroaspis venom to be more similar to a Naja venom than to a Crotalus venom. There are logically distinctive differences between venoms of Elapid and Vipera genera, and future studies could focus on them, but the SVM in this initial study was defined so that it would differentiate between two families. This aim was reached. Future studies will aim at narrowing this down to genera level and possibly even further.
Reviewer 2 – Comment 14) Also, specify which species the venom came from. What was their age, etc.? Venom varies with age, habitat, sex, and diet within species.
ANSWER: As mentioned above as reply to previous comments, unfortunately this data is not available for our venom set. We did change the following text in our manuscript in lines 531-535: “This study has certain limitations that allow for bias to potentially modify the results: as it has been explained, factors such as sex or age -that have not been considered for the model building in this study - might interfere with the obtained results.” into “This study has certain limitations that allow for bias to potentially modify the results. For example, factors such as sex, age and/or geographical location have not been considered for the model building in this study as this data is not available for a large part the venoms in our venom collection and/or some venom samples are from pooled milkings of different individuals.” To further specify the issue with disclosing these properties of the snakes
Reviewer 2 – Comment 15) Line 606: Explain internal standards.
ANSWER: The reviewer might have missed the explanation on the internal standards. This is elaborated on in lines 138-161: “To check for the repeatability of the analyses, the venom of Naja siamensis was spiked with a mixture of standards -specified in the Materials and Methods section- and analyzed several times before the analysis of all the venoms was performed. As can be seen in Figure S.1 – Analysis of repeatability of the Supporting Information, the retention times of the features were sufficiently repeatable for the bioinformatic tool to extract the features and to recognize the same ones amongst the different repetitions -even if the retention times shift slightly-, meaning that the method was repeatable. The standard’s mixture added to this venom was used to investigate whether normalization of the intensity of all found features by an internal standard would render more reproducible results. Thus, the intensities of the signals of these standards were used to normalize the extracted intensities from the found features by means of recalculating all the intensities as a percentage relative to the chosen standard’s signal intensity, for all included standards separately. The highest repeatability was found when the extracted intensities were not normalized. This is probably because all intensities were normalized using a single standard at a time, and internal standards should preferably be similar to the compounds that are being standardized, which was impossible for this study since none of the compounds measured were known at the point of measurement and of data processing. Also, taking into account the complexity of snake venom, no analyte can be similar to all of the found metabolites, meaning that no single internal standard could be used. Thus, no standardization was performed on the data. However, one of the standards (d9-Caffeine) was included in all the measurements to check for any issues during the analyses. By monitoring this standard in all measurements, the stability of the analyses could be investigated, from which it was found that the measurements were performed robustly.”
Reviewer 2 – Comment 16) Line 602-606: Needs citation
ANSWER: Only one study has been performed on snake venom metabolites where a mixture of two standards was used -also without citation (cited as [8] in the manuscript). We reached to our set of standards by testing 30 different compounds at different concentrations, checking whether they provided UV absorption at the two measured wavelengths, and whether they would suffer from ion suppression or not. The 5 internal standards that were used in the end were the only compounds that appropriately followed these parameters. Furthermore, our standards are only used to check for the reproducibility of the analytical measurements, for which many different compounds can be chosen.
Reviewer 2 – Comment 17) Line 606-607: Not needed.
ANSWER: We believe that in the Experimental section of papers it is required to tell from which supplier chemicals came from, as it benefits reproducibility. In this case Sigma Aldrich, which we mention.
Reviewer 2 – Comment 18) Line 621: Why did the author(s) select Naja Siamensis? There needs to be an explanation.
ANSWER: We thank the reviewer for this comment. This was also raised in a previous question by the same reviewer in Comment 10), where we addressed it.
Reviewer 2 – Comment 19) Line 630: Please delete the main part of this study.
ANSWER: We thank the reviewer for this comment and now address it as follows: “For the main part of this study, a flow splitter [...].” was changed into “For all LC-MS analyses, a flow splitter […]”
Reviewer 2 – Comment 20) Section 5.2.3: Is missing necessary details but has extraneous details. Please rectify.
ANSWER: This has already been addressed in our ANSWER of comment 12 of the same reviewer.
Reviewer 3 Report
Dear Authors;
I reviewed your manuscript submitted as toxins-2131767, and I found it very interesting with a large amount of scientific results.
There are minor corrections that are better to be realized before publishing:
1. Line 602: It was stated that “50 μL of 1 mg/mL solutions of each venom were injected into the HPLC-MS”, but in Table S.2: Order of the injections in the Supporting Information, the concentration was indicated as “5 mg/mL”, which one is correct?
2. I suggest that the model that are have used in statistical techniques, for better understanding to be summarized in one flowchart.
3. Fig. 5a: The numbers and letters inside the plot are not clear, which are recommended to be corrected.
Best Regards
Round 2
Reviewer 2 Report
It's interesting research and quite important. My main concern is that the authors have stated in the abstract that "this study aids in the development of tools to fight Tropical Disease Snakebite." However, they have not demonstrated in the paper how this study helps in combating "snakebite" as an issue. I would recommend that the authors not say this study helps in combating the "snakebite issue" or specify specifically how this study helps in combating the issue.
Author Response
ANSWER: We agree with the reviewer’s comment, and we have thus deleted the mentioned lines [16-17].